# Removal of Fluoride from Aqueous Solution Using Shrimp Shell Residue as a Biosorbent after Astaxanthin Recovery

**DOI:** 10.3390/molecules28093897

**Published:** 2023-05-05

**Authors:** Yan Li, Lili Zhang, Minru Liao, Chao Huang, Jing Gao

**Affiliations:** 1Collage of Food Science, Guangdong Pharmaceutical University, Zhongshan 528458, China; 2Collage of Food Science and Technology, Guangdong Ocean University, Zhanjiang 524088, China

**Keywords:** fluoride, adsorption, shrimp shell, waste, ionic liquid, astaxanthin

## Abstract

Natural astaxanthin has been widely used in the food, cosmetic, and medicine industries due to its exceptional biological activity. Shrimp shell is one of the primary natural biological sources of astaxanthin. However, after astaxanthin recovery, there is still a lot of chitin contained in the residues. In this study, the residue from shrimp (*Penaeus vannamei*) shells after astaxanthin extraction using ionic liquid (IL) 1-ethyl-3-methyl-imidazolium acetate ([Emim]Ac) was used as a bioadsorbent to remove fluoride from the aqueous solution. The results show the IL extraction conditions, including the solid/liquid ratio, temperature, time, and particle size, all played important roles in the removal of fluoride by the shrimp shell residue. The shrimp shells treated using [Emim]Ac at 100 °C for 2 h exhibited an obvious porous structure, and the porosity showed a positive linear correlation with defluorination (*D_F_*, %). Moreover, the adsorption process of fluoride was nonspontaneous and endothermic, which fits well with both the pseudo-second-order and Langmuir models. The maximum adsorption capacity calculated according to the Langmuir model is 3.29 mg/g, which is better than most bioadsorbents. This study provides a low-cost and efficient method for the preparation of adsorbents from shrimp processing waste to remove fluoride from wastewater.

## 1. Introduction

Fluoride is beneficial for bones and dental enamel development [1]. However, excessive fluoride in drinking water may cause bone poisoning, osteoporosis, and a weakened immune system [2]. Thus, removing excessive fluorides from water is still an urgent task. Various technologies, such as electrocoagulation [3], ion exchange [4], and membrane techniques [5], have attracted significant interest in fluoride removal. Nevertheless, the high operating cost or the use of toxic chemicals have restricted their applications in water treatment [2].

Adsorption techniques have been extensively used to remove impurities and pollutants from wastewater because of their low maintenance cost, easy operation, and simple design [6,7]. *Schima wallichii*-activated carbon [8] and Ce-Ti@Fe_3_O_4_ nanoparticles [9] both showed an efficient removal of fluoride with a maximum adsorption capacity of 2.524 and 94.01 mg/g, respectively. In addition, porous alumina hollow spheres expressed good adsorption of fluoride when the calcination temperature was higher than 800 °C [10]. Recently, researchers have been working on developing low-cost and natural adsorbents. Bioadsorbents, such as orange peel cellulose [11], waste ginkgo shells [12], and chicken eggshell powder [13], were considered as potential adsorbents for water treatment. Using agricultural wastes as adsorbents can not only alleviate environmental pressure but also improve the economic value of the by-products [14].

During shrimp processing, plenty of heads, shells, and tails are removed, which account for 40~60% of the raw material weight. Many methods have been explored to recover high-value molecules (chitin, protein, ash, and astaxanthin) from the by-products of shrimp processing [14]. However, these residues still contain a lot of chitin, which still causes resource waste. Previously, chitin and chitosan were demonstrated as absorbents to remove dyes (such as RB5) [15], heavy metals (Pb, Fe, Cu, etc.) [16,17], and antibiotics [18]. However, the traditional methods for the preparation of chitin and chitosan from shrimp shells include deproteinization, deacetylation, hydrothermal carbonization, and acid washing, which are very complex and environmentally harmful.

Ionic liquids (ILs) have unique properties, such as a low melting point, high chemical and thermal stability, strong solubility, and designability [19,20,21]. ILs and IL solutions have been widely used as substitutions for organic solvents to extract astaxanthin from shrimp shells [14,22]. Nevertheless, the structure of the shrimp shell residue after astaxanthin recovery was ignored in the previous literature. In fact, shrimp shells are ideal raw materials as bioadsorbents because their structure is hierarchical [23,24,25]. However, the surface of native shrimp shells displays a smooth, dense, and flat morphology due to a high protein content and some mineral salts.

In this work, shrimp shell residue from astaxanthin recovery was applied as an adsorbent to remove fluoride from water. To acquire the optimal bioadsorption and understand the adsorption process, the effect of the particle size, solid–liquid ratio, and the extraction temperature and time on the chemical component, structure, and defluorination (*D_F_*, %) of the [Emim]Ac-treated shrimp shells (ETSSs) were analyzed. Moreover, *D_F_* under different adsorption conditions, such as the initial fluoride concentration, adsorbent doses, and pH were investigated to enhance the adsorption efficiency of the ETSSs. Furthermore, the adsorption modeling was evaluated to reveal the adsorption mechanism.

## 2. Results and Discussion

### 2.1. Characterization of ETSSs

#### 2.1.1. Component Analysis

Shrimp shells generally consist of compact matrices of chitin fibers interlaced with calcium carbonate and proteins [14]. The hierarchical structure of the raw shrimp shells could be observed at the beginning of the dissolution (Appendix A). During the dissolution process (100 °C for 2 h), a large amount of chitin fibers were isolated from the shrimp shells and dispersed in [Emim]Ac (Appendix A).

To understand the [Emim]Ac treatment for shrimp shells, the main components before and after the dissolution under various conditions (time, temperature, and solid–solvent ratio) were analyzed, and the data are listed in Table 1. In general, the contents of ash and proteins of the ETSSs were lower than raw shrimp shells, indicating ash and proteins could be removed using [Emim]Ac. It was demonstrated in the previous literature that almost all of the calcium carbonate and proteins could be dissolved in an ammonium-based IL [24]. However, the removal efficiency of ash is higher than that of proteins in this study, which may be attributed to the tight combination between proteins and chitin [26].

#### 2.1.2. XRD Analysis

The crystal structure of the ETSSs was studied using XRD patterns, and the diffraction pattern of all the samples displays two remarkable diffraction peaks around 2*θ* angles of 9.0° and 20.0° (Appendix A), which can be attributed to glucosamine sequences and *N*-acetyl-ᴅ-glucosamine monomers, respectively [27]. The crystallinity was calculated based on the intensity of the crystalline region and amorphous region, and the CrI data are shown in Table 1. The CrI values of the ETSSs were all lower than the raw shrimp shells due to the removal of the calcium carbonate/calcium carbonate crystal (main composition of ash) from the shrimp shells [28,29]. The reduction of CrI for the ETSSs was in accord with the decrease in the content of ash. During astaxanthin extraction, the [Emim]Ac treatment at 60–120 °C for 1–3 h could partly remove the ash and proteins, which may have resulted in the structural change of the shrimp shells.

#### 2.1.3. SEM Analysis

The surface morphology of the shrimp shells before and after treatment with [Emim]Ac was compared, and the SEM images are shown in (Figure 1). In general, the raw shrimp shells exhibited a smooth and close-knit layered structure, while the ETSS samples showed rough and loose structures. Moreover, the larger particle size (>180 μm) of the shrimp shells and [Emim]Ac dosage are beneficial to form a porous surface. However, a long time (>3 h) and high temperature (>120 °C) may lead to chitin degradation and agglomerate formation. The shrimp shells with a particle size of 180–250 μm pretreated using [Emim]Ac at 100 °C for 2 h displayed a porous structure of interconnected micropores formed between irregular particles. The pore size and density distribution were basically uniform. The micropore sizes ranged from 0.2 to 0.4 μm, and the porosity was 238.4 vol.%, indicative of a high surface area (Table 1). After treatment, the single-layer panel structure for shrimp shells changed to a sponge-like construction, which could be attributed to the removal of calcium carbonate and proteins. It should be stressed that our purpose is not to extract chitin by completely removing proteins and ash from shrimp shells but to provide a simple method to prepare a bioadsorbent that has a strong adsorption capability.

### 2.2. Fluoride Removal Using ETSSs

After treatment, ETSSs were applied to adsorb fluoride in wastewater, and the *D_F_* values are listed in Table 1. The results show that the highest *D_F_* was 36.36% when the shrimp shells with a particle size of 180–250 μm were pretreated at 100 °C for 2 h at a solid/liquid ratio of 1:10 *w*/*w*, while *D_F_* was only 5.23% when the untreated shrimp shells were used as the bioadsorbent. However, prolonged treatment time or increased treatment temperature are not conducive to absorbing fluoride using ETSSs.

To further evaluate the effect of [Emim]Ac treatment on fluoride removal, the relationship between the composition, crystal structure, surface characteristics of ETSSs, and *D_F_* was investigated. As shown in Figure 2, *D_F_* of the ETSSs increased with the decrease in ash content, protein content, and CrI value while decreasing with the decrease in porosity. Moreover, there were good linear relationships between *D_F_* and the ash content and protein content, as well as porosity (*R*^2^ > 0.80). Similarly, Mohan et al. [30] showed that the number of pores on the surface of chitin improves the ability of chitin to adsorb metal ions. This situation supported that the adsorption process of fluoride was related to the porous structure of ETSSs resulting from [Emim]Ac treatment. Therefore, the ETSSs with the maximum chitin content (57.10%) and porosity (238.40%) were further used in the adsorption experiment.

### 2.3. Optimization of Adsorption Conditions

#### 2.3.1. Effect of Initial Fluoride Concentration

For a given adsorbent dose (10 g/L), the effect of the initial fluorine concentration on fluorine removal was studied. Figure 3a shows *D_F_* decreasing with the increase in initial fluoride concentration, which was consistent with the previous report of defluorination using brewery waste diatomite [31]. Generally speaking, the higher the fluoride ion content, the greater the concentration gradient at the solid–liquid interface, thereby increasing the adsorption [32]. Although the presence of more fluoride enhanced the utilization of adsorption sites at the beginning, the capacity of the adsorbent was exhausted with increasing fluoride. Therefore, the studied ETSSs have a good fluoride removal efficiency (>90%) in high-fluoride-containing water (10–20 mg/L). For example, with an initial fluoride concentration of 10 mg/L, *D_F_* could reach up to 94.64%, and the residual fluoride (*R_F_*, mg/L) was only 0.51 mg/L, which is much lower than the limitation of the WHO (<1.5 mg/L).

#### 2.3.2. Effect of Adsorbent Dose

As shown in Figure 3b, *D_F_* increased from 50.36% to 98.33% with an increase in ETSS doses from 5 g/L to 25 g/L. Similarly, the removal efficiency of Pb(II)/Cd(II) increased with the increasing amount of magnetic chitin [16]. With the increase in ETSS dosage, the increase in removal efficiency is mainly due to the increase in the number of effective fluorine-absorbing active sites [31]. However, the increase in *D_F_* is not obvious due to a lower utilization efficiency of ETSSs once the adsorbent dose exceeds 15 g/L. Therefore, in order to save adsorbents and increase the *D_F_* value, the adsorbent dose of 15 g/L should be given priority.

#### 2.3.3. Effect of pH

Solution pH plays a predominant role in interacting with fluoride and an adsorbent. Fluoride adsorption was evaluated over a pH range of 3–10 (Figure 3c). The results show that the fluoride sorption escalated as the pH decreased from 7 to 3. Similar results have also been reported that acidic media are more favorable for removing excess fluoride from water [33]. At an acidic pH, the electronegative functional groups (such as -OH, C=O, and -NH) on the surface were protonated, and they were most suitable for binding with anions [11,34]. At a lower pH, more cations on the surface of the ETSSs led to greater adsorption of the fluoride.

On the other hand, *D_F_* reached an equilibrium state from pH 7 to pH 10. It may be because the number of positively charged sites on the adsorbent surface decreases while increasing the solution pH [35]. In an alkaline system, the surface of ETSSs was accumulated with negative charge OH^-^ ions, causing repulsion between the negatively charged surface and fluoride. On the contrary, Dan et al. [36] reported that the acid-treated *Moringa oleifera* leaves-based adsorbent showed a maximum *D_F_* of 83% at pH 1, whereas the maximum *D_F_* of the alkali-treated *Moringa oleifera* leaves-based adsorbent was 85% at pH 10. Similarly, Dobaradaran et al. [37] found that the *D_F_* of raw shrimp shells increased with increasing pH, and an alkaline environment was more efficient. The fluoride ions may form strong bonds with the various positive charges on the surface of different bioadsorbents.

As a result, the surface of the ETSSs is positively charged in an acidic environment, and the negatively charged F^−^ ions are electrostatically attracted to be removed from the water. Figure 3c shows a maximum *D_F_* of 85% at pH 3. Therefore, ETSSs show great potential for removing fluoride from acidic wastewater.

### 2.4. Adsorption Kinetics

The reaction time has a significant effect on the adsorption process, and the fluoride adsorption kinetic curve is shown in Figure 4a. It can be seen that the uptake rate of fluoride on ETSSs significantly increased in the first 1 h due to a large number of available adsorption sites for fluoride adsorption. The increase in adsorption rate was relatively slow from the adsorption time of 2 h, and the adsorption process finally reached an equilibrium at 8 h with an equilibrium adsorption capacity of 3.13 mg/g.

In order to verify the kinetic mechanisms of fluoride adsorption on the ETSSs, pseudo-first-order and pseudo-second-order kinetic models were selected to evaluate the type and order of the adsorption process [38]. The linear fitting equations and parameters are shown in Table 2, and the fitting curve of the pseudo-first-order and pseudo-second-order kinetic models are presented in Figure 4b and Figure 4c, respectively. Both the pseudo-first-order (*R*^2^ = 0.9985) and the pseudo-second-order models (*R*^2^ = 0.9987) showed good linear correlation coefficients with *R*^2^ > 0.99. Moreover, the (equilibrium adsorption uptake) *q_e_* calculated using the pseudo-second-order model (*q_e_* = 3.400 mg/g) is very close to the experimental value (*q_e_* = 3.130 mg/g), while the *q_e_* calculated by the pseudo-first-order model (*q_e_* = 1.620 mg/g) differs greatly. Therefore, the pseudo-second-order model is more suitable for describing fluoride adsorption on ETSSs, indicating that fluoride adsorption on ETSSs is a typical chemisorption by sharing covalent bonds or the exchange of electrons between fluoride and the ETSSs [39]. Some of the earlier studies also reported the aptness of pseudo-second-order kinetics for fluoride adsorption, such as *Moringa oleifera* leaves-based biosorbents [36], coconut root-based biosorbents [40], Ficus Glomerata Bark-based biosorbents [41], and banana peel-based biosorbents [42].

### 2.5. Adsorption Isotherm

Four different isotherm models, including Langmuir, Freundlich, Temkin, and Dubinin–Radushkevich (D-R), were used to predict the adsorption capacity of the ETSSs and the mechanism of the adsorption process [43]. The fitting equations and parameters are summarized in Table 3. Figure 5 shows the fitting curve of the four isotherm models with the experimental adsorption equilibrium data. All isotherm models exhibited good agreement with the experimental values, especially at low *C_e_*. Moreover, the *q_e_* tended to equilibrium when the *C_e_* exceeded 18 mg/L, and the fluoride adsorption on the ETSSs was limited.

As shown in Table 3, the Langmuir model based on a monolayer of solute molecules on the adsorbent surface fitted well (*R*^2^ = 0.9970) with the experimental data. In other words, the adsorption of fluoride on the ETSSs was a monolayer adsorption process. Therefore, the increased ETSS surface area facilitated fluoride adsorption, and the ETSSs with a porosity of 238.40% obtained the maximum *D_F_* (36.36%) (Table 1). Moreover, the *q_e_* predicted by the Langmuir model (*q_e_* = 3.290 mg/g) is larger than the experimental (*q_e_* = 3.13 mg/g) one but very close. For the Langmuir model, the equilibrium parameter *R_L_* is described as 1/(1 + *K_L_C*_0_) by Swain et al. [35]. The value of *R_L_* < 1 represents favorable adsorption, while an unfavorable adsorption is if *R_L_* > 1. In this study, the values of *R_L_* are 0.02–0.11, indicating that the adsorption of fluoride using ETSSs is favorable.

On the other hand, the Freundlich model with a poor fitting coefficient (*R*^2^ = 0.8964) indicated that the process of fluoride adsorption on the ETSSs was not multilayer adsorption. Furthermore, the good fitting relationships between the Temkin model (*R*^2^ = 0.9803) and the experimental data revealed a uniform distribution of ETSS surface binding energy in the process of fluoride adsorption. In addition, the D-R model with *R*^2^ = 0.9317 was also favorable for the adsorption of fluoride on the ETSSs. According to the assumption of the D-R model [35], fluoride tends to fill the pores on the inhomogeneity surface of ETSSs during the adsorption process.

Therefore, the adsorption of fluoride on ETSSs is a combination of chemical adsorption (monolayer adsorption) and physical absorption (porous mosaic). This adsorption characteristic of ETSSs is significantly different from that of raw shrimp shells as bioadsorbents. Gok et al. [44] showed that the most suitable isotherm models for the adsorption of Co^2+^ by raw shrimp shells were Langmuir (*R*^2^ = 0.9567) and Freundlich (*R*^2^ = 0.9387) rather than the D-R model (*R*^2^ = 0.7032). That is to say, only chemisorption occurs when raw shrimp shells are used as bioadsorbents, which may be related to the smooth surface morphology of raw shrimp shells. Pure chitin with a smooth surface morphology exhibits similar adsorption behaviors to raw shrimp shells [45]. Low *D_F_* (<50%, adsorbent dose = 10 g/L) was obtained using chitin as a bioadsorbent, according to Kamble et al. [46]. In conclusion, ETSSs treated using ionic liquid can be used as the preferred adsorbent for fluoride adsorption because of the synergistic effect of chemical and physical adsorption.

### 2.6. Thermodynamic Parameters

To evaluate the thermodynamic behavior of the adsorption of fluoride by ETSSs the adsorption experiments were performed at different temperatures. As shown in Figure 6, increasing adsorption benefits the adsorption capacity, and the optimum temperature for fluoride adsorption is 60 °C with an adsorption capacity of 3.32 mg/g. The thermodynamic parameters were determined to understand and authenticate the sorption nature. The thermodynamic parameters, including Gibb’s free energy (∆*G*), positive enthalpy (∆*H*^0^), and positive entropy (∆*S*^0^) were investigated, and the results are presented in Figure 6. The results of ∆*G* are all positive, indicating the adsorption process of fluoride with ETSSs is nonspontaneous. In a previous study, fluoride adsorption using prawns derived as the adsorbents was also nonspontaneous [33]. The ∆*H*^0^ demonstrates the adsorption process is endothermic, indicating that the increase in temperature favors the adsorption process [33]. The results were in accord with the trend shown in Figure 6. The ∆*S*^0^ indicates that fluoride has a good affinity with ETSSs and the randomness of the ETSS/solution interface increases during the adsorption process.

### 2.7. Adsorption Mechanism

After [Emim]Ac treatment at 100 °C for 2 h, the shrimp shell residue exhibited loose and porous structural properties by removing proteins and calcium. In the adsorption process, fluoride first overcomes the water resistance and reaches the ETSS surface and pores. Subsequently, the increased surface area is available for sharing or exchanging the fluoride with the hydroxyl groups in chitin. To prove the theory, the pH value of the wastewater before and after defluoridation was investigated. As expected, the pH of the solution increased from 3.01 to 8.17 after adsorption. Thus, it could be expected that fluoride adsorption using the ETSSs as the adsorbent was mainly dominated by chemisorption and can be described as follows:
[C_8_H_13_NO_3_(OH)_2_]_n_ (s) + 2nF^−^ (aq) → [C_8_H_13_NO_3_F_2_]n (s) + 2nOH^−^ (aq)

A similar reaction mechanism for the adsorption of fluoride on typical activated carbon [47] or cashew nut shell carbon-based biosorbents [48] was also suggested.

## 3. Materials and Methods

### 3.1. Materials

Shrimp (*Penaeus Vannamei*) shells were collected from Zhanjiang Guolian Aquatic Products Co. Ltd. (Zhanjiang, Guangdong, China). The materials were washed thoroughly with flowing tap water and dried for 6 h in a hot air oven at 90 °C. The dried material was ground in a mill and passed through screens of different particle sizes (120–250 µm). [Emim]Ac (99%) was provided by Lanzhou Institute of Chemical Physics. The purity of other reagents used in this experiment is analytically pure.

### 3.2. Bioadsorbent Preparation

Dried shrimp shells and [Emim]Ac were mixed at the mass ratio of 1:1–1:10 g/mL in a 50 mL round flask. The mixture was continuously stirred with a magnetic stirrer (Model DF-101S, Henan Yuhua instrument company, Zhengzhou, China). The treatment conditions including solid–solvent ratio, time, and temperature are listed in Table 1. The dissolving processes were observed at 50 × using an Olympus-BH-2 microscope. After treatment, the mixed solution was regenerated using deionized water and centrifuged at 8000× *g* for 10 min. The bioadsorbent was obtained after drying the solid production at 60 °C for 8 h.

### 3.3. Characterization

The moisture content of shrimp shells and ETSSs was examined using a halogen moisture meter (HX204, Mettle-Toledo Company, Switzerland) at 105 °C. The protein content was calculated using Kjeldahl’s method [49]. The determination of the ash content was based on the method from Mohan et al. [30]. A scanning electron microscope (SEM) (Japan JSM 6510-EDAX) and an alpha radiation spectrometer were used to characterize the surface morphology of shrimp shells and ETSSs under an accelerating voltage of 20 kV. Images were obtained at magnification of 2000×. X-ray diffraction (XRD) measurements were used to evaluate the crystalline structure of the bioadsorbent. The ETSSs were tested using Bruker D8 Advance at 40 kV and 30 mA voltage. The samples were scanned continuously over 2*θ* angles ranging from 5° to 45°. The crystalline index value (CrI) was calculated according to Equation (1) [35]:(1)$$\mathrm{CrI}(\%)=\frac{I_{110}-I_{am}}{I_{110}}\times 100$$
where *I*_110_ is the maximum diffraction intensity at 2*θ* = 20°, and *I_am_* is the amorphous diffraction intensity at 2*θ* = 16°.

### 3.4. Adsorption Experiment

For the adsorption screening, 10 mL of different concentrations (10–60 mg/L) and pH (3–10) of fluoride solution were mixed with a desired content of ETSSs in a 50 mL round bottom flask. The solid–liquid ratio was 1:5, 1:10, 1:20, and 1:25 g/mL. After blending, the mixture was stirred using a magnetic stirrer at 20–60 °C for 1–10 h. The mixed sample was quantitatively aspirated through a pipette every hour and then centrifuged at 8000× *g* for 10 min to separate the ETSSs from the liquid. Subsequently, the fluoride ion concentration in the supernatant was determined using the selective fluoride ion method (Model PF-1-01, Shanghai Yiliang Scientific Instruments Company, Shanghai, China) [50]. The equilibrium adsorption uptake (*q_e_*, mg/g), *D_F_*, and residual fluoride (*R_F_*, mg/L) were calculated as follows:(2)$$q_{e}=\frac{(C_{0}-C_{e})\times V}{m}$$
(3)$$R_{F}^{}=C_{0}-C_{e}$$
(4)$$D_{F}=\frac{C_{0}-C_{e}}{C_{0}}\times 100$$
where *m* (g) is the dose of adsorbent; *C*_0_ (mg/L) and *C_e_* (mg/L) are the concentration of fluoride at initial time and adsorption equilibrium, respectively; and *V* (L) is the volume of NaF solution.

Adsorption kinetic data were modeled using pseudo-first-order and pseudo-second-order models, and the linear fitting equations are as follows [11,38]:
(5)$$\begin{array}{ll} \mathrm{Pseudo-first-order} & \log(q_{e}-q_{t})=\log q_{e}-\frac{k_{1}}{2.303}t \end{array}$$
(6)$$\begin{array}{ll} \mathrm{Pseudo-second-order} & \frac{t}{q_{t}}=\frac{1}{k_{2}q_{e}{}^{2}}+\frac{t}{q_{e}} \end{array}$$
where *k*_1_ (min^−1^) and *k*_2_ (g∙min/mg) are the rate constants of the pseudo-first-order and pseudo-second-order models, respectively, and *q_t_* is the amount of fluoride adsorbed at time *t*.

Application of adsorption isotherm models can help to design optimum adsorption conditions of adsorbents for engineering applications. Therefore, the Langmuir, Freundlich, Temkin, and Dubinin–Radushkevich (D-R) models were used to substantiate the fluoride adsorption behavior of the ETSSs.

The Langmuir isotherm based on a monolayer of solute molecules on the adsorbent surface is expressed as shown by Equation (7) [43]:(7)$$q_{e}=\frac{q_{m}K_{L}C_{e}}{1+K_{L}C_{e}}$$
where *q_m_* (mg/g) is the maximum monolayer adsorption capacity, and *K_L_* (L/mg) is the adsorption constant of Langmuir.

The Freundlich isotherm gives an empirical relationship based on heterogeneous surface sorption and multilayer adsorption phenomenon [43]. The linear form of Freundlich isotherm is defined as follows:(8)$$q_{e}=K_{F}C_{e}{}^{1/n}$$
where *K_F_* (mg/g) is the adsorption constant of Langmuir, and *n* is the adsorption intensity.

The Temkin isotherm characterized by uniform distribution of surface binding energy is fitted by Equation (9) [31]:(9)$$q_{e}=K_{T}\ln f+K_{T}\ln C_{e}$$
where *K_T_* is the adsorption constant of Temkin model, and *f* (L/mg) is Temkin binding constant.

The D-R adsorption isotherm is often used to explain the inhomogeneity of the solid surface energy in the microporous monolayer region [35]. The isotherm constant (*β*) and mean sorption energy (*E*) were determined using Equations (10)–(12): (10)$$\varepsilon =RT\ln(1+\frac{1}{C_{e}})$$
(11)$$\ln q_{e}=\ln q_{m}-\beta \varepsilon^{2}$$
(12)$$E=\frac{1}{2\beta^{0.5}}$$
where *ε* (mol^2^/kJ^2^) is the Polanyi potential; *β* (mol^2^/kJ) is the activity coefficient related to mean sorption energy; and *E* (kJ/mol) is the sorption energy.

The thermodynamic parameter of Gibb’s free energy change (∆*G*^0^), enthalpy change (∆*H*^0^), and entropy change (∆*S*^0^) were calculated using Equations (13) and (14) [49]:
(13)$$\Delta G^{0}=-RT\ln(\frac{q_{e}}{C_{e}})$$
(14)$$\ln(\frac{q_{e}}{C_{e}})=\frac{\Delta S^{0}}{R}-\frac{\Delta H^{0}}{RT}$$
where ∆*G*^0^ (kJ/mol), ∆*H*^0^ (kJ/mol), and ∆*S*^0^ (kJ /mol/K) are the free energy change, the enthalpy change, and entropy change, respectively; *R* (8.314 J/mol/K) is the universal gas constant; and *T* (K) is absolute temperature.

### 3.5. Recovery of IL

[Emim]Ac was recovered using aqueous biphasic systems (ABSs) according to the previous work [50]. After the treatment, the supernatant, mainly containing [Emim]Ac, was separated, and a certain amount of K_3_PO_4_ was then added into the liquid to form ABS. The mixture was shaken using a vortex mixer (XW-80A, Jingke, Inc., Shanghai, China) and allowed to stand overnight at 25 °C. Both phases were carefully separated, and the [Emim]Ac content in upper phase was quantified via HPLC (LC-20A, Prominence, Tokyo, Japan). Each sample was diluted using varying ratios in water before injection. The experimental design of fluoride adsorption using ETSSs is shown in Figure 7.

### 3.6. Data Analysis

All the results were expressed as mean ± standard deviation. Origin Pro 8.0 software was used for correlation and regression analysis. ChemDraw 16.0 and Chem3D 16.0 were used to draw the molecular structures and mechanism diagrams.

## 4. Conclusions

In this work, shrimp shells were treated using [Emim]Ac and then used as bioadsorbents to remove fluoride ions from wastewater. [Emim]Ac treatment at 100 °C for 2 h resulted in a loose and porous structure of the adsorbents with an average micropore size of 0.3 and porosity of 238.4 vol.%. The enhanced surface area was beneficial to improve fluoride removal, and the porosity showed a positive linear correlation with *D*_F_. Throughout the kinetics studies, the adsorption fitted well with the pseudo-second-order model. From isotherm studies, the Langmuir adsorption model fitted the experimental data, proving that the mechanism of fluoride adsorption onto ETSSs occurs via chemisorption with strong interaction of the electrostatic forces between fluoride ions and hydroxyl groups of chitin in ETSSs. From the thermodynamics analysis, adsorption was identified as feasible and endothermic and occurred nonspontaneous. After adsorption, the *R_F_* of the wastewater was reduced from 50 mg/L to 0.83 mg/L. The adsorption capacity of fluoride using different bioadsorbents was compared with ETSSs, and the data are listed in Table 4. It is obvious that the maximum adsorption capacity calculated by the Langmuir model is 3.29 mg/g, which is better than the vast majority of reported bioadsorbents. The results demonstrate that the shrimp shell residue from astaxanthin recovery using ILs can be used as a bioadsorbent for water treatment.

## Figures and Tables

**Figure 1 molecules-28-03897-f001:**
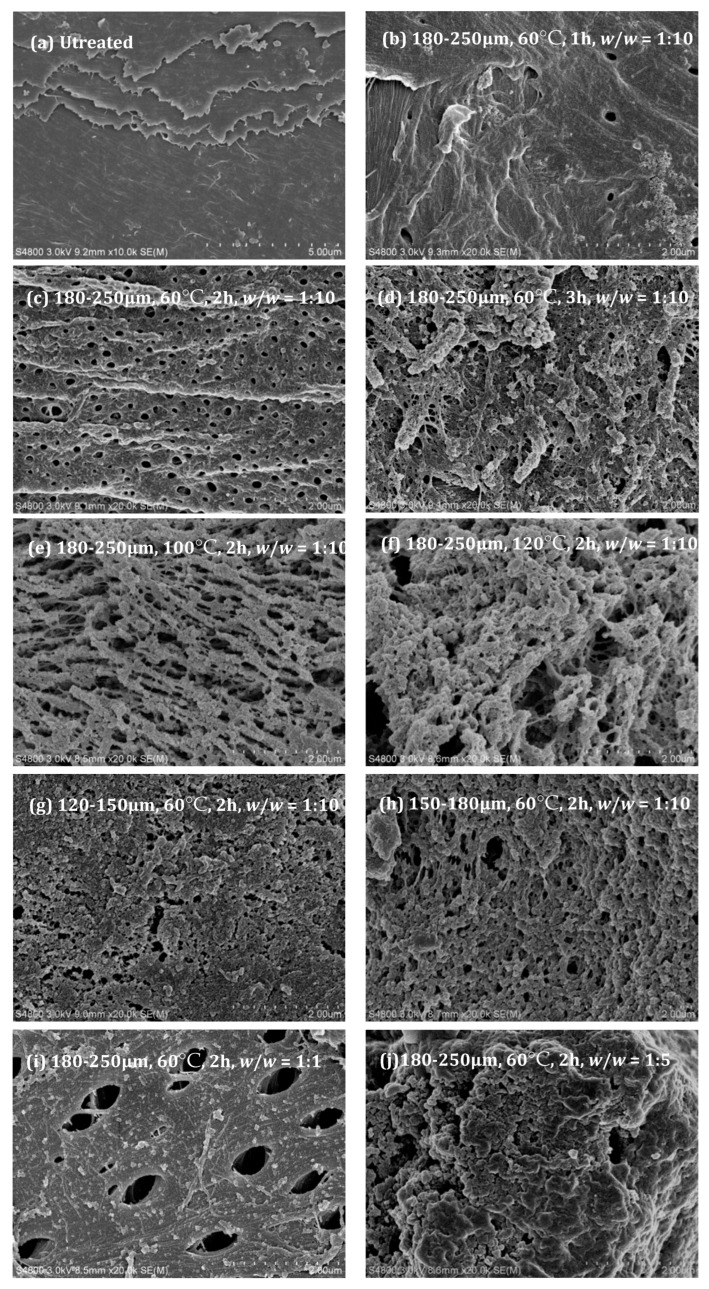
SEM images of ETSSs treated under different conditions.

**Figure 2 molecules-28-03897-f002:**
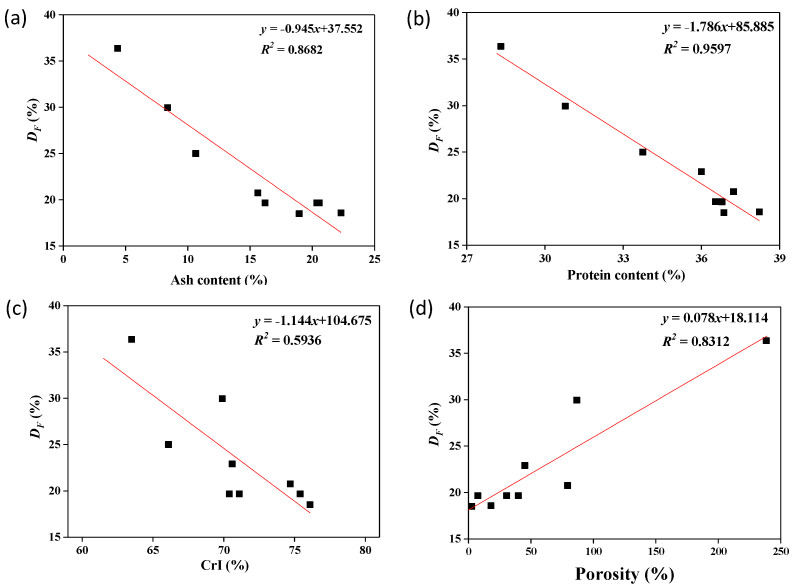
*D_F_* value as a function of ash content (**a**), protein content (**b**), CrI (**c**), and porosity (**d**) of the ETSSs.

**Figure 3 molecules-28-03897-f003:**
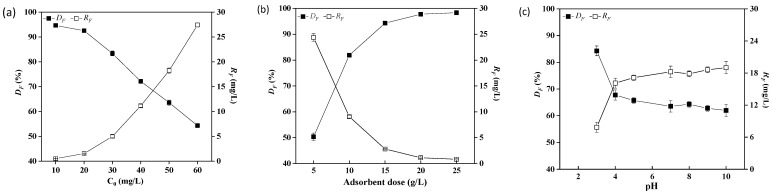
Effect of initial fluoride concentration *C*_0_, adsorbent dose, and pH on *D_F_* and *R_F_*: (**a**) *w*/*v* = 10 g/L, 50 °C, 8 h, initial pH = 6.3; (**b**) *C*_0_ = 50 mg/L, 50 °C, 8 h, pH = 3; (**c**) *C*_0_ = 50 mg/L, *w*/*v* = 10 g/L, 50 °C, 8 h.

**Figure 4 molecules-28-03897-f004:**
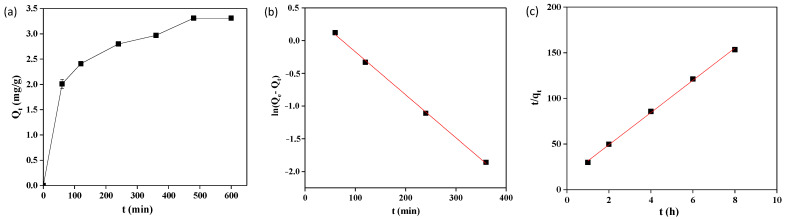
Kinetic data for fluoride adsorption of ETSSs (pH = 7, *T* = 30 °C, adsorbent dose = 10 g/L, and *C*_0_ = 50 mg/L): (**a**) adsorption amounts of ETSSs with various contact times; (**b**) pseudo-first-order kinetic model; (**c**) pseudo-second-order kinetic model.

**Figure 5 molecules-28-03897-f005:**
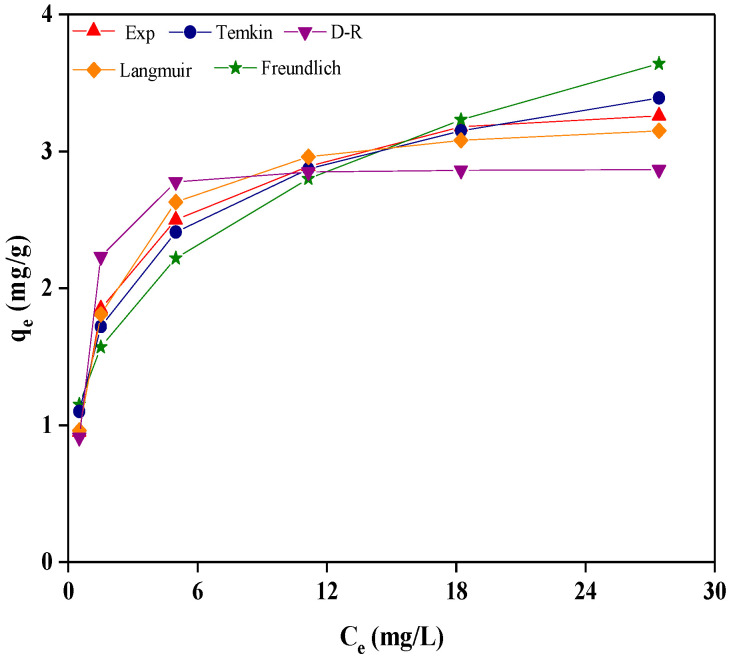
Adsorption isotherm models for the removal of fluoride using ETSSs (pH = 7, *T* = 50 °C, adsorbent dose = 10 g/L, *C*_0_ = 10–60 mg/L, and *t* = 8 h).

**Figure 6 molecules-28-03897-f006:**
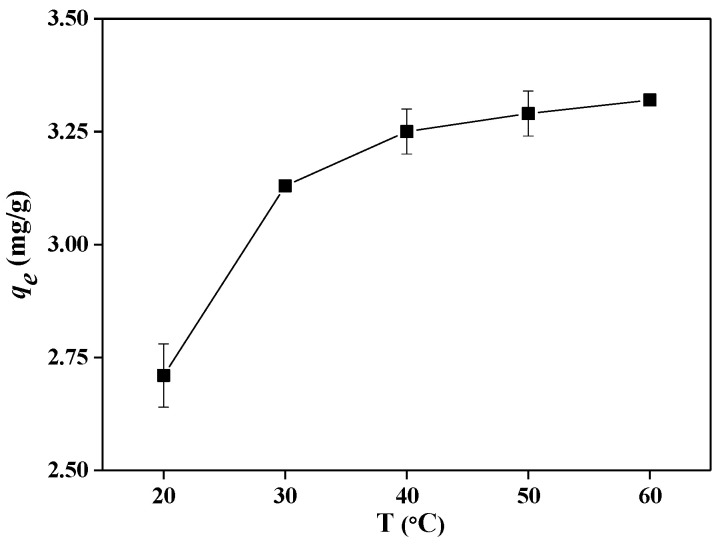
Adsorption capacity of fluoride as a function of temperature (pH = 7, adsorbent dose = 10 g/L, *C*_0_ = 50 mg/L, and *t* = 8 h).

**Figure 7 molecules-28-03897-f007:**
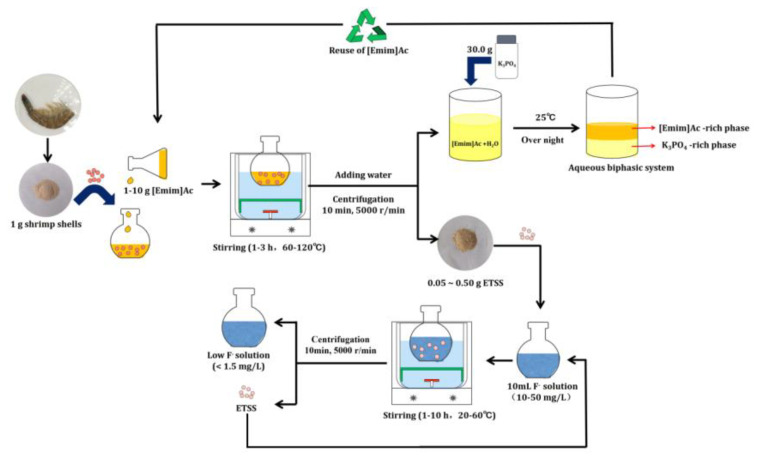
The experimental design of fluoride adsorption using ETSSs.

**Table 1 molecules-28-03897-t001:** The chemical compositions, structure properties, and *D_F_* of shrimp shell residue treated using [Emim]Ac under different conditions.

Treatment Conditions				Chemical Compositions			Structure Properties				
Solid–Liquid Ratio (*w*/*v*)	Time (h)	Particle Size (μm)	Temperature (°C)	Ash (%)	Protein (%)	Chitin (%)	MN ^a^ (30.76 μm^2^)	APD ^b^ (nm)	Porosity (%)	CrI (%)	*D_F_* ^c^ (%)
-	-	180–250	-	28.50 ± 0.06	42.90 ± 0.35	22.74 ± 0.05	-	-	-	81.8	5.23 ± 0.02
1:10	1	180–250	60	18.95 ± 0.14	36.86 ± 0.33	9.70 ± 0.10	10	157.78	2.54	76.1	18.58 ± 0.05
1:10	2	180–250	60	14.38 ± 0.05	36.00 ± 0.36	39.70 ± 0.45	296	122.18	45.11	70.6	22.90 ± 1.82
1:10	3	180–250	60	16.21 ± 0.18	32.31 ± 0.32	43.57 ± 0.44	648	77.78	40.02	70.4	19.67 ± 1.85
1:10	2	180–250	100	4.36 ± 0.06	28.31 ± 0.32	57.10 ± 0.25	352	257.58	238.40	63.5	36.36 ± 0.10
1:10	2	180–250	120	8.38 ± 0.08	30.78 ± 0.26	51.02 ± 0.37	20	651.85	86.75	69.9	29.95 ± 1.67
1:10	2	120–150	60	22.31 ± 0.07	38.22 ± 0.20	30.56 ± 0.23	309	75.56	18.01	54.7	18.58 ± 0.26
1:10	2	150–180	60	20.36 ± 0.03	36.54 ± 0.32	38.77 ± 0.31	340	93.94	30.63	71.1	19.67 ± 1.89
1:5	2	180–250	60	15.63 ± 0.09	37.23 ± 0.34	38.16 ± 0.68	228	184.67	79.37	74.7	20.75 ± 1.85
1:1	2	180–250	60	20.57 ± 0.11	36.79 ± 0.25	36.22 ± 0.23	57	113.33	7.47	75.4	19.66 ± 1.80

Note: the first row are the results for raw shrimp shells. ^a^ MN = microporous number; ^b^ APD = average pore diameter; ^c^ initial fluoride concentration (*C*_0_) = 50 mg/L, initial pH (6.3), adsorbent dose = 10 g/L, adsorption time = 2 h, and temperature = 30 °C.

**Table 2 molecules-28-03897-t002:** Kinetic parameters of the pseudo-first-order and pseudo-second-order kinetic models for fluoride adsorption using ETSSs ^a^.

Experiment Value	Pseudo-First-Order	Pseudo-Second-Order
*q_e,exp_* = 3.13 mg/g	*k*_1_ = 0.39 min^−1^	*k*_2_ = 0.006 g∙mg^−1^∙min
	*q_e,cal_* = 1.62 mg/g	*q_e,cal_* = 3.40 mg/g
	*R*^2^ = 0.9985	*R*^2^ = 0.9987

Note: ^a^ pH = 7, T = 30 °C, adsorbent dose = 10 g/L, and *C*_0_ = 50 mg/L.

**Table 3 molecules-28-03897-t003:** The isotherm models and parameters for fluoride adsorption using ETSSs ^a^.

Isotherm Model	Langmuir	Freundlich	Temkin	D-R
Fitting parameter	*K_L_* = 0.805 L/mg	*K_F_* = 1.394 mg/g	*K_T_* = 0.576	*E* = 1.924 kJ/mol
	*q_m,cal_* = 3.290 mg/g	1/*n* = 0.290	*f* = 13.147 L/mg	*q_m,cal_* = 2.869 mg/g
*R* ^2^	0.9970	0.8964	0.9803	0.9317

Note: ^a^ pH = 7, *T* = 50 °C, adsorbent dose = 10 g/L, *C*_0_ = 10–60 mg/L, and *t* = 8 h.

**Table 4 molecules-28-03897-t004:** Comparison of characteristic of the proposed method with some published methods for fluoride removal based on Langmuir isotherm.

Bioadsorbent	Defluoridation Conditions				*q_e,max_* (mg/g)	Reference
	Time (h)	Temperature (°C)	*C*_0_ (mg/L)	Dose (g/L)		
Zirconium-modified pea peel waste carbon	1.0	25	0–50	-	3.652	[35]
Brewery waste diatomite	0.5	25	-	60	0.617	[31]
*Ficus benghalensis* leaf	1.5	27	2–25	8	2.242	[51]
Coconut-shell-derived carbon nanotube	3.5	30	2.68–9.57	10	0.360	[52]
Aluminum-modified activated carbon from Khat waste	1.0	Room temperature	2–9	2.47	0.306	[53]
Wattle humus biosorbent	-	30	2–10	-	0.231	[54]
Modified *Moringa oleifera* leaves	2.5	Room temperature	0.5–2	2.5	1.14	[36]
CaCl_2_-modified *Crocus sativus* leaves	-	25	2–25	10	2.01	[55]
Ce(IV)-modified orange juice residue	24	30	0–12	0.67	1.22	[56]
Shrimp shells treated using [Emim]Ac	8	50	10–60	10	3.290	This study

Note: “-” means not mentioned in the literature.

## Data Availability

Not applicable.

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
