# Peer review of "Removal of Fluoride from Aqueous Solution Using Shrimp Shell Residue as a Biosorbent after Astaxanthin Recovery"

_molecules, 2023, doi:10.3390/molecules28093897_

Round 1
Reviewer 1 Report
The manuscript submitted by authors related to the “Removal of Fluoride from Aqueous Solution using Shrimp Shells Residue as a biosorbent after Astaxanthin Recovery” is an interesting work. In the introduction authors described good the problem. But still manuscript lacks in several ways. This manuscript needs major revision concerning several things. The comments are provided below for the author’s attention. Please make the writing more rigorous and scientific addressing the following comments.
The manuscript is written on a very old template (See the header 2021).
The conclusion of the abstract states that “This study provides a low-cost and efficiency method for preparation adsorbent from shrimp processing waste to remove fluoride, metal ion, and chemical dyes from wastewater” but this study was not conducted with metal ions and chemical dyes.
In the section on the effect of pH, for clarity, you can describe detail the effect of pH on the adsorption of fluoride.
In the introduction line 40-42 please include a recently published article, https://doi.org/10.3390/su15054470 (2023) bioadsorbent for the removal of pollutants from water.
In the introduction line 48-49 include a very recent reference to the removal of pollutants using shrimp shells from water. https://doi.org/10.3390/su15032431 (2023).
The method section is so short. Results and discussion contain methods. Please separate it.
Describe details of sampling including the time duration of sampling.
Figure 1(d). Please write correctly 180-250 um.
Figure 2 could be subdivided as a, b, c, d, and the legend also accordingly.
Figure 3 has no a, b, or c but the legend describes a, b, c. Please revise the figure.
Point 11: The reusability and stability of the adsorbent should be checked.
There are so many typo grammatical errors in the whole manuscript, which should be revised by some native speaker and formatting should be checked. Some examples: please correct Line 351, ……..the data are listed in…
Line 287, ………The dried material was ground…
Line 289, complete sentence… [Emim] Ac 288 (99%), provided by Lanzhou Institute of Chemical Physics.
Line 292…correct the information g/g (maybe g/mL or w/v).
Line 297 …centrifuged at 8000 r/min for 10 min… It can be written in × g.
It is suggested to include a spectrum of SEM-EDX with fluoride could be included along with other inorganic elements.
It is suggested to include several more results in Table 4. that compares the uptake performance of fluoride by similar materials vs. this adsorbent.
I am not a native but I found several mistakes. Some of them are mentioned in my comments.
Author Response
- Comments:The manuscript is written on a very old template (See the header 2021).
Reply: We feel sorry for our carelessness. In our resubmitted manuscript, the template is revised. Thanks for your correction.
- Comments: The conclusion of the abstract states that “This study provides a low-cost and efficiency method for preparation adsorbent from shrimp processing waste to remove fluoride, metal ion, and chemical dyes from wastewater” but this study was not conducted with metal ions and chemical dyes.
Reply: We are agree with your suggestion. The statement “This study provides a low-cost and efficiency method for preparation adsorbent from shrimp processing waste to remove fluoride, metal ion, and chemical dyes from wastewater” have been corrected to “This study provides a low-cost and efficiency method for preparation adsorbent from shrimp processing waste to remove fluoride from wastewater”.
- Comments: In the section on the effect of pH, for clarity, you can describe detail the effect of pH on the adsorption of fluoride.
Reply: According for your kind suggestion, we have described detail the effect of pH on the adsorption of fluoride in section “2.3.3 Effect of pH”.
- Comments: In the introduction line 40-42 please include a recently published article, https://doi.org/10.3390/su15054470(2023) bioadsorbent for the removal of pollutants from water.
Reply: As suggested by the reviewer, we have added more references in the introduction part. The article entitled “Investigation of Efficient Adsorption of Toxic Heavy Metals (Chromium, Lead, Cadmium) from Aquatic Environment Using Orange Peel Cellulose as Adsorbent” in Sustainability (2023) gives new ideas for the application of Agro-wastes in water treatment. We have cited this article (ref. 11) in the Introduction line 41 and therefore the order of the references were adjusted.
- Comments: In the introduction line 48-49 include a very recent reference to the removal of pollutants using shrimp shells from water.https://doi.org/10.3390/su15032431 (2023).
Reply: Thanks for the suggestion. A potential candidate, chitosan, extracted and isolated from shrimp shells was provided in the study. This article (ref. 17) has been cited in the Introduction line 50.
- Comments: The method sectionis so short. Results and discussion contain methods. Please separate it.
Reply: We thank the reviewer for pointing out this issue, we have restructured the method section in the revised manuscript. Some methods mentioned at the “results and discussion” have been moved to “Section 3 Materials and methods”. Consequently, the descriptions of adsorption kinetics, isotherm and thermodynamic parameters have been moved to section 3.4.
- Comments: Describe details of sampling including the time duration of sampling.
Reply: Thanks for the suggestion, and the detailed statement has been added in the method (3.4. Adsorption experiment).
- Comments: Figure 1(d). Please write correctly 180-250 um.
Reply: Figure 1(d). have been corrected.
- Comments: Figure 2 could be subdivided as a, b, c, d, and the legend also accordingly.
Reply: Figure 2 have been subdivided as a, b, c, d, and the legend also accordingly.
- Comments: Figure 3 has no a, b, or c but the legend describes a, b, c. Pleaserevise the figure.
Reply: We have revise the Figure 3. Figure 3 have been subdivided as a, b,
- Comments: The reusability and stability of the adsorbent should be checked.
Reply: The reusability and stability of the adsorbent are common concern of many researchers. Nevertheless, we did not conducted this experiment for the following reasons. On the one hand, researchers have observed that the reusability of bioadsorbents based on chemisorption is poor. For example, Angelin et al (https://doi.org/10.1007/s11356-021-14864-9, 2021) reported that the fluoride removal efficiency of the regenerated wattle humus adsorbent was drastically reduced around 60 to 70% while increasing the initial fluoride concentration from 6 to 10 mg/L, although the fluoride removal efficiency was retained even at fifth cycle. On the other hand, the process of adsorbent regeneration may consume much organic solvents, strong acid or strong base (such as HCl, NaOH, formic acid, et al) (http://dx.doi.org/10.1016/j.jece.2017.04.013, 2017), which is not in line with the concept of green development.
- Comments: There are so many typo grammatical errors in the whole manuscript, which should be revised by some native speaker and formatting should be checked. Some examples: please correct Line 351, ……..the data are listed in…
Line 287, ………The dried material was ground…
Line 289, complete sentence… [Emim] Ac 288 (99%), provided by Lanzhou Institute of Chemical Physics.
Line 292…correct the information g/g (maybe g/mL or w/v).
Line 297 …centrifuged at 8000 r/min for 10 min… It can be written in × g.
Reply: Thanks for the suggestion. All the mistakes have been carefully corrected and the language has been improved in the revised manuscript.
- Comments: It is suggested to include a spectrum of SEM-EDX with fluoride could be included along with other inorganic elements.
Reply: We thank the reviewer for the suggestion. In the study, the pH of the solution increased from 3.01 to 8.17 after the fluoride adsorption by the [Emim]Ac-treated shrimp shells. Combining the results of adsorption kinetics and isotherm, it could be expected that the ion exchange existed between fluorineand the [Emim]Ac-treated shrimp shells.
- Comments:It is suggested to include several more results in Table 4. that compares the uptake performance of fluoride by similar materials vs. this adsorbent.
Reply: According to your suggestion, we have add more results in Table 4. that compares the uptake performance of fluoride by similar materials vs. this adsorbent.
Reviewer 2 Report
The paper "Removal of Fluoride from Aqueous Solution using Shrimp Shells Residue as a biosorbent after Astaxanthin Recovery" presents useful and interesting information on an alternative fluoride adsorbent. However, it needs some improvements before further consideration. Here are my comments and suggestions:
1. Line 61 – There is an error in the sentence “In this work, shrimp shells residue from astaxanthin recovery was applied as adsorption to remove fluoride from water” – “adsorption” should be corrected to “adsorbent”.
2. In Table 1, please clearly indicate that in the first row are the results for raw shrimp shells.
3. Please specify under what adsorption conditions the DF values presented in Table 1 were obtained.
4. As the XRD data are discussed, please provide some example of XRD patterns (raw and treated shrimp shells) in the Supplementary material.
5. In Figure 1, the scale is faintly visible, please correct.
6. Were SEM and XRD analyses also performed after fluoride adsorption? The authors should include such analyses for the adsorbent with the highest removal efficiency and discuss changes (if any) before and after fluoride adsorption.
7. For the section Effect of initial fluoride concentration (line 136) please specify a range of concentration.
8. What was the reason for conducting all adsorption experiments at adsorbent dose of 10 g/L since the Authors clearly stated that the adsorbent dose of 15g/L should be given priority.
9. A similar question to the temperature at which adsorption was performed - isotherm and kinetic studies were carried out at 30 °C while other (effect of individual parameters) at 50 °C.
10. In the caption of Figure 5 and the footnote to Table 3, please correct the initial concentration of 50 mg/L to the appropriate concentration range.
11. Please specify in the section Material and methods for which adsorbent the study of the effect of various parameters on the adsorption process was carried out - in the Abstract the shrimp shells after treatment at 60 °C are mentioned, and in the section 2.7 (Adsorption mechanism, line 271) the shrimp shells after treatment at 100 °C.
12. Have adsorbent regeneration tests been carried out? If the Authors claim (Fig. 7) that the adsorbent is suitable for reuse, it should be supported by appropriate results.
13. In table 4, the Authors reported that the temperature and initial concentrations in this study were 50 °C and 10-60 mg/L, while earlier in the manuscript other process conditions were given. Please, check.
Author Response
- Comments: Line 61 – There is an error inthe sentence “In this work, shrimp shells residue from astaxanthin recovery was applied as adsorption to remove fluoride from water” – “adsorption” should be corrected to “adsorbent”.
Reply: The sentence “In this work, shrimp shells residue from astaxanthin recovery was applied as adsorption to remove fluoride from water” – “adsorption” have been corrected to “adsorbent”.
- Comments: In Table 1, please clearly indicate that in the first row are the results for raw shrimp shells.
Reply: We have clearly indicated that in the first row are the results for raw shrimp shells at the end of the Table 1.
- Comments: Please specify under what adsorption conditions the DFvalues presented in Table 1 were obtained.
Reply: We have specified the adsorption conditions of the DF values presented in Table 1 at the end of the Table 1.
- Comments: As the XRD data are discussed, please provide some example of XRD patterns (raw and treated shrimp shells) in the Supplementary material.
Reply: XRD patterns have been provided in the Supplementary material.
- Comments: In Figure 1, the scale is faintly visible, please correct.
Reply: The scale in Figure 1 was corrected in the revised manuscript.
- Comments: Were SEM and XRD analyses also performed after fluoride adsorption? The authors should include such analyses for the adsorbent with the highest removal efficiency and discuss changes (if any) before and after fluoride adsorption.
Reply: SEM and XRD showed the structural properties of shrimp shells before and after astaxanthin recovery at different extraction conditions. The analyse for the adsorbent with the highest removal efficiency was stressed in the manuscript (2.2. Fluoride removal by ETSS). The changes before and after fluoride adsorption will be investigated in another study.
- Comments: For the section Effect of initial fluoride concentration (line 136) please specify a range of concentration.
Reply: The defluorination of the [Emim]Ac-treated shrimp shells decreased with the increase of initial fluoride concentration. As shown in Fig. 3a, the studied [Emim]Ac-treated shrimp shells exhibited a good fluoride removal efficiency ( > 90%) when the experiment was carried out under the conditions of w/v = 10 g/L, 50℃, 8 h, and initial pH (6.3). The initial fluoride concentration of 10-20 mg/mL have been specified in the revised manuscript.
- Comments: What was the reason for conducting all adsorption experiments at adsorbent dose of 10 g/L since the Authors clearly stated that the adsorbent dose of 15g/L should be given priority.
Reply: According to previous research experience (https://doi.org/10.1002/clen. 201800286, 2019; https://doi.org/10.1080/00986445. 2018. 1423969, 2018 ), the adsorbent content was initially fixed at 10 g/L. Therefore, adsorbent dose of 10 g/L was used in adsorption experiments. However, the adsorbent dose of 15g/L showed better efficiency in the subsequent study of the effect of adsorbent dose on fluoride removal efficiency. The two results do not contradict each other.
- Comments: A similar question to the temperature at which adsorption was performed - isotherm and kinetic studieswere carried out at 30 °C while other (effect of individual parameters) at 50 °C.
Reply: We are grateful for the question. Similarly, the effect of adsorption time on the defluorination was first studied in the optimized experiment and a temperature of 30℃ (near room temperature) was selected in the kinetic studies. Subsequently, the effect of temperature on the defluorination was investigated. Although the highest defluorination (DF, %) was obtained at 60℃ (DF = 66.34%, qe = 3.32 mg/g), 50℃ (DF = 65.74%, qe = 3.29 mg/g) was chosen as the most suitable adsorption temperature because of high defluorination, low energy consumption and mild to protein.
- Comments: In the caption of Figure 5 and the footnote to Table 3, please correct the initial concentration of 50 mg/L to the appropriate concentration range.
Reply: We have made correction in the in the revised manuscript according to your suggestion.
- Comments: Please specify in the section Material and methods for which adsorbent the study of the effect of various parameters on the adsorption process was carried out - in the Abstract the shrimp shells after treatment at 60 °C are mentioned, and in the section 2.7 (Adsorption mechanism, line 271) the shrimp shells after treatment at 100 °C.
Reply: We are very sorry for our incorrect writing in the Abstract. In the Abstract, the statement “the shrimp shells after treatment at 60℃” have been corrected to “the shrimp shells after treatment at 100℃” .
- Comments: Have adsorbent regeneration tests been carried out? If the Authors claim (Fig. 7) that the adsorbent is suitable for reuse, it should be supported by appropriate results.
Reply: Thanks for the suggestion. The adsorbent regeneration tests have not been carried out, thus the Fig. 7 has been revised.
- Comments: In table 4, the Authors reported that the temperature and initial concentrations in this study were 50 °C and 10-60 mg/L, while earlier in the manuscript other process conditions were given. Please, check.
Reply: We are very sorry, and have carefully checked the experimental conditions in the revised manuscript.
Round 2
Reviewer 1 Report
Dear authors,
Thank you for responding to most of the comments. The revised manuscript is better now.
Author Response
Thanks very much for your suggestion.
Reviewer 2 Report
The authors have revised the manuscript appropriately based on the comments from the reviewer. However, there are still two minor issues that the Authors should pay attention to:
1. The vertical axis signature Rf is missing after the separation of Figure 2, please check.
2. In Figure 4, the horizontal axis scale is not completely visible.
With the exception of these two remarks, the current version of this manuscript is ready for publication.
Author Response
1. The vertical axis signature Rf is missing after the separation of Figure 2, please check.
Author's Notes: Thanks for the suggestion. We checked Figure 2, and provided PDF of the revised version. The defluorination (DF, %) can be seen in Figure 2.
2. In Figure 4, the horizontal axis scale is not completely visible.
Author's Notes: We are very sorry for the mistake. Figure 4 has been checked and revised. The horizontal axis scale can be seen completely visible.